# Antifungal Effect of Autochthonous Aromatic Plant Extracts on Two Mycotoxigenic Strains of *Aspergillus flavus*

**DOI:** 10.3390/foods12091821

**Published:** 2023-04-27

**Authors:** Francisco Ramiro Boy, Rocío Casquete, Iris Gudiño, Almudena V. Merchán, Belén Peromingo, María José Benito

**Affiliations:** 1Nutrición y Bromatología, Escuela de Ingenierías Agrarias, Universidad de Extremadura, Avd. Adolfo Suárez s/n, 06007 Badajoz, Spain; ramboy1@hotmail.com (F.R.B.); igudino@unex.es (I.G.); avmerchan@unex.es (A.V.M.); belenperomingo@unex.es (B.P.); mjbenito@unex.es (M.J.B.); 2Instituto Universitario de Investigación en Recursos Agrarios (INURA), Universidad de Extremadura, Avd. de la Investigación, 06006 Badajoz, Spain

**Keywords:** aromatic Dehesa plants, phenolic compounds, antifungal activity, *Aspergillus flavus*, aflatoxins

## Abstract

This study identified the compounds obtained from four native Dehesa plants, which were holm oak, elm, blackberry and white rockrose, and evaluated their ability to inhibit the growth and production of aflatoxins B_1_ and B_2_ of two strains of mycotoxigenic *Aspergillus flavus*. For this purpose, phenolic compounds present in the leaves and flowers of the plants were extracted and identified, and subsequently, the effect on the growth of *A. flavus*, aflatoxin production and the expression of a gene related to its synthesis were studied. *Cistus albidus* was the plant with the highest concentration of phenolic compounds, followed by *Quercus ilex*. Phenolic acids and flavonoids were mainly identified, and there was great variability among plant extracts in terms of the type and quantity of compounds. Concentrated and diluted extracts were used for each individual plant. The influence on mold growth was not very significant for any of the extracts. However, those obtained from plants of the genus *Quercus ilex*, followed by *Ulmus* sp., were very useful for inhibiting the production of aflatoxin B_1_ and B_2_ produced by the two strains of *A. flavus*. Expression studies of the gene involved in the aflatoxin synthesis pathway did not prove to be effective. The results indicated that using these new natural antifungal compounds from the Dehesa for aflatoxin production inhibition would be desirable, promoting respect for the environment by avoiding the use of chemical fungicides. However, further studies are needed to determine whether the specific phenolic compounds responsible for the antifungal activity of *Quercus ilex* and *Ulmus* sp. produce the antifungal activity in pure form, as well as to verify the action mechanism of these compounds.

## 1. Introduction

Aromatic plants are a group of plants whose active compounds are constituted, totally or partially, by essences, representing about 0.7% of the total number of medicinal plants [1]. Their extracts have been used for various purposes, as well as in medicine, food, cosmetics, pharmacy and perfumery due to their strong antioxidant activity since they contain polyphenols, vitamin E, selenium, vitamin C, β-carotene, lutein, lycopene, and other carotenoids that play a fundamental role in adsorbing and neutralizing free radicals to improve human health [2], slowing the development of diseases related to oxidative stress such as cancer, diabetes or Alzheimer’s disease. These plants are considered a future option, especially for areas with special qualities (mountainous, scrubland, moderate arid areas, etc.) where conventional crops are not feasible or profitable [3].

The Extremadura Dehesa is a region in Spain that has been declared a UNESCO Biosphere Reserve and is characterized as a very diverse ecosystem. A great heterogeneity of aromatic and medicinal plants can be found [4]. This included *Quercus ilex*, whose extracts have been highlighted for their anti-inflammatory, antibacterial, hepatoprotective, antidiabetic, anticancer, gastroprotective, antioxidant and cytotoxic properties [5]. Other Dehesa plants such as *Ulmus* sp., *Rubus ulmifolius* and *Cistus albidus* have been studied, and from their extracts, vitamins (ascorbic acid), reducing sugars (glucose and fructose), polyunsaturated fatty acids and eicosadienoic acids have been obtained [6]. The functional properties offered by these plants correspond mainly to the phenolic compounds found in their fruits, seeds, leaves, stems and flowers. These constitute a relatively wide group of compounds that belong to the secondary metabolism of plants and whose molecular structure presents a hydroxyl group linked to an aromatic ring, which confers antimicrobial capacity. They are organized into several groups; phenolic acids, flavonoids, and non-flavonoids are the main groups [7]. Several studies have demonstrated the antimicrobial effects of these phenolic compounds, highlighting their use against pathogenic bacteria and spoilage yeasts in food [4]. Consequently, these plants could be studied as well as an alternative to avoid and reduce the presence of molds in food and, consequently, the occurrence of mycotoxins.

Mold growth in food may cause different food spoilage and serious health risks for consumers due to the production of toxic metabolites. The genera *Penicillium*, *Aspergillus*, *Fusarium*, *Alternaria*, *Cladosporium*, *Mucor* and *Rhizopus* are included in this group [8]. The mycotoxins that are produced have toxic effects even when consumed in small amounts since they have accumulative effects on the organism [9]. Several EU regulations currently exist on maximum limits allowed for the most widely recognized mycotoxins, aflatoxin, ochratoxin, zearalenone fumonisin and patulin. These regulations are particularly severe in terms of the limits required for foods and food products destined for infants and children, as they are more susceptible to the effects of mycotoxin contamination [10]. Among the mycotoxins, aflatoxins, produced mainly by *Aspergillus flavus*, are undoubtedly the most documented of all mycotoxins and are widely present in foods [11]. There are six types of aflatoxins (B_1_, B_2_, G_1_, G_2_, M_1_ and M_2_) [12], where aflatoxin B_1_, AFB_1_, stands out for being considered as Class I carcinogens by the World Health Organization (WHO). More specifically, the bioaccessibility of AFB_1_ in infant formulas has been reported to be higher than for other mycotoxins, being, therefore, more toxic [10]. They can be both acutely and chronically toxic and be carcinogenic, mutagenic, teratogenic and immunosuppressive to most mammalian species [13,14]. Aflatoxins have been detected in many foods, but their presence is especially important in maize, cottonseeds, peanuts and other nuts [15,16]. Conventional chemical and physical methods are insufficient to remove toxins from foods, and the use of synthetic compounds to inhibit mold growth can lead to resistance and health problems [17]. For these reasons, an urgent need exists to identify a healthy alternative strategy to reduce aflatoxin contamination by molds using natural compounds that can affect inhibiting mold growth or block biosynthesis. In the case of pathways directly related to aflatoxin biosynthesis, the *aflR* gene is the regulatory gene for aflatoxin synthesis, and it has been described that there is an intimate correlation between the expression of this gene and the production of the aflatoxins AFB_1_ and AFG_1_ [18,19,20].

This study aims to evaluate the compounds present in the native plants of the Dehesa, holm oak, elm, blackberry and white rockrose, identifying the compounds present in each extract and subsequently assaying their antifungal activities against two strains of mycotoxigenic *A. flavus*, as well as their influence on aflatoxin AFB_1_ and AFB_2_ production and expression of genes related to these mycotoxins.

## 2. Materials and Methods

### 2.1. Plant Material

Four aromatic plants have been used in this work, the leaves of *Quercus ilex* (Holm oak), *Ulmus* sp. (Elm), *Rubus ulmifolius* (Blackberry) and the flowers of *Cistus albidus* (White rockrose) provided by a company located in a Dehesa area of Peraleda de la Mata municipality, Cáceres, Extremadura. They were collected in February 2021. All samples were packed in vacuum bags and stored at 4 °C until use.

### 2.2. Fungal Strains

For this study, two strains of the aflatoxigenic mold *A. flavus*, Cq103 and Cq8, that produce aflatoxin B1 and B2 from the collection of the CAMIALI group from the University of Extremadura, Spain, were used. Both were seeded in a Petri dish in Potato Dextrose Agar (PDA; Oxoid, Madrid, Spain) and allowed to grow at 25 °C for five days [19].

### 2.3. Extraction of Phenolic Compounds from Plants

For the extraction of phenolic compounds from plants, the ultrasonic extraction method was used [4]. The resulting aqueous extracts were mixed to a known final volume.

The extracts were purified using HYPERSEP C18 500MG/3ML/50PLG purification columns, through which 10 mL of methanol and 10 mL of Milli-Q water were passed, followed by a vacuum to activate them, and then air was allowed to pass. Finally, 2 mL of the sample extracts were passed through and washed with 10 mL of Milli-Q water, and the sample was collected in a test tube with methanol. The solution was then evaporated in a rotary evaporator under a vacuum at 35 °C and resuspended in water.

### 2.4. Determination of Total Phenolic Content and Phenolic Compound Identification

The Folin–Ciocalteu method described by Wettasinghe and Shahidi [21] was used to determine the total phenolic content. For this purpose, 0.5 mL of sample, 10 mL of Milli-Q water and 1 mL of Folin–Ciocalteu’s reagent was added to a 25 mL volumetric flask, shaken and waited for about 3 min, then 2 mL of saturated sodium carbonate was added, shaken rapidly, the flask was made up to volume with Milli-Q water and left for 1 h in the dark. Finally, the absorbance of the samples was measured at 760 nm. Gallic acid was used as standard. Results are expressed as mg gallic acid equivalents (GAE)/100 g of fresh plants. All experiments were performed in triplicate. The bioactive compounds identification was performed by HPLC-UV-ESI-MS/MS with the extracts diluted to 100 mg/L in methanol. Samples were filtered through a 0.22 μm syringe filter and injected into an HPLC (Agilent HPLC-QTOF Model G6530, Agilent Technologies, Palo Alto, CA, USA) equipped with a C18 column (4.6 × 1.50 mm, × 4.8 µm) Agilent Technologies. Phenolic compounds identification was performed with a quadrupole time-of-flight (Q-TOF) tandem mass analyzer with an electrospray ionization (ESI) source. The gas flow rate was 11 mL/min at 280 °C (35 psi nebulizer). Gradient elution was done with a mixture of hydrocyanic acid/water (5:95, *v*/*v*) as solvent A and hydrocyanic acid/water/formic acid (95:4.9:0.1 *v*/*v*/*v*) as solvent B, with a flow rate of 0.350 mL/min. The solvent gradient started with solvent B/solvent A (5:95, *v*/*v*), reaching 90:10 (*v*/*v*) at 15 and 20 min and returning to initial conditions during the last 10 min. The tentative identification of bioactive compounds was elucidated from the MassBank database. All samples were injected in triplicate.

### 2.5. Antifungal Experimental Conditions and Sampling

Once the molds were grown on PDA agar plates, the spores were collected with 2 mL of sterile Milli-Q water using 1 mL micropipettes and transferred to a sterile Eppendorf tube. Neubauer chamber was used to quantify spore suspension and adjusted to 10^6^ spores/mL. This procedure was performed in a safety room and Bunsen burner. PDA plates were inoculated with 100 µL of different concentrations of each extract (250 and 150 mg/L) spread with the aid of a glass loop and 10 µL of the spore suspension of each mold. The strains were inoculated separately. Inoculation of the suspension of each mold, Cq8 and CQ103, were used, and the negative control was done with water. The experiment was carried out for 11 days, during which the Petri dishes were incubated at 25 °C in order to evaluate the reaction of the aflatoxigenic molds at their optimum growth temperature. The experiment was carried out in triplicate.

### 2.6. Mold Growth Monitoring

To evaluate the effect of the purified extracts on the growth of both strains of *A. flavus*, the colony diameter (mm) was measured in two directions at right angles to each other every day at the same time to prevent alterations in the measurements following the methodology specified by Casquete et al. [22]. Next, regression curves, growth (mm) versus days of incubation, were obtained for the linear phase of mold growth under study. Generally, mold growth was considered to have a linear phase when the correlation coefficient of the line (R2) was very close to 1. Data that did not fit a linear regression of the line were rejected in the construction of the line. Next, the growth rate of *A. flavus* in the presence and absence of the purified plant extracts was determined. The experiment was carried out in triplicate.

### 2.7. Extraction and Quantification of Aflatoxins

For extraction and quantification of aflatoxins was done at the end of the assay. Half of the contents of the Petri dishes were poured into a Stomacher bag individually, 100 mL of chloroform was added in a fume hood, mixed well, and allowed to settle for 1 h in the dark. The solution was filtered in a rotary evaporator flask with a filter to which two tablespoons of anhydrous sulfate were added. Next, a wash was made by adding 10 mL of chloroform to the bag. The filtrate was brought to evaporate at a rotary evaporator at a temperature of 30 °C; once evaporated, 2 mL of chloroform was added and transferred to Kimax tubes to evaporate the solvent in the Mivac with nitrogen gas. Then, 1 mL of chloroform was added to these tubes and evaporated again. Next, 500 μL of methanol and 500 μL of Milli-Q water were added and shaken in a vortex, passing the solution into a vial through a 0.2 μM filter with a 1 mL syringe. For aflatoxin quantification, vial samples were analyzed on HPLC with a diode array detector [18]. The mobile phase used for the separation contained HPLC-grade water (solvent A) and HPLC-grade acetonitrile (solvent B), run in a gradient mode set from 15% B at the start to 100% B at 30 min. Three minutes were necessary to equilibrate the column. All solvents used were purchased from Thermo Fisher Scientific (Runcorn, UK). The injection volume was 25 μL, and the flow rate was 1 mL/min. Calibrations were carried out for each aflatoxin, B_1_ and B_2_, using standards obtained from Sigma–Aldrich.

### 2.8. Gene Expression Studies

For the study of the relative expression of the *aflR* gene [19] involved in aflatoxin synthesis, the other half of the plate was scraped with a scalpel and frozen at −80 °C until RNA extraction. RNA was isolated using the Plant Total RNA Kit (Sigma–Aldrich, St. Louis, MO, USA). Samples of 250 to 500 mg of mycelium biomass were ground to powder in a pre-frozen mortar with liquid nitrogen, and total RNA was purified. Gene expression studies were performed by using reverse-transcription real-time quantitative PCR. First, cDNA was synthesized using about 500 ng of total RNA according to the kit protocol (Takara Bio Inc., Otsu, Shiga, Japan) and then used for quantitative PCR analysis. Primer pairs AflRTaq1/2 (AflRTaq1 TCGTCCTTATCGTTCTCAAGG; AflRTaq2 ACTGTTGCTACAGCTGCCACT) were used to amplify the regulatory gene (*aflR*) and the primer pair F/R-TUBjd (F-TUBjd TCTTCATGGTTGGCTTCGCT; R-TUBjd CTTGGGGTCGAACATCTGCT) amplified the β-tubulin (β-TUB.JD) gene, used as the endogenous control [19]. The qPCR reactions were carried out using an Applied Biosystems 7300 Fast Real-Time PCR system with SYBR Premix Ex Taq (Takara Bio Inc.) in the PCR reaction. The expression of the *aflR* gene with the housekeeping gene β-tubulin as an endogenous expressed control was quantified [20].

### 2.9. Statistical Analysis

Statistical analysis of the data was carried out using SPSS for Windows, version 21.0 (IBM Corp., Armonk, NY, USA). Descriptive statistics of the data were determined, and the differences within and between groups were studied by one-way analysis of variance (ANOVA) and separated by Tukey’s honestly significant differences test (*p* ≤ 0.05). Principal component analysis (PCA) was performed on the correlation matrix of the variables.

## 3. Results and Discussion

### 3.1. Quantification and Identification of Phenolic Compounds Extracted from Plants

The results of the extracts’ concentration obtained by ultrasound from each type of plant are presented in Table 1. The table shows that *Quercus ilex* had twice the concentration of phenolic compounds compared to *Rubus ulmifolius* and *Ulmus* sp., but the plant with the highest concentration of compounds was *Cistus albidus*, with 703 mg AGE/100 g of sample.

Considering that for the first three plants, the leaves were used and are trees, while for *Cistus albidus*, which is a shrub, the flowers were used, it is reasonable to assume that in the flowers, there is a greater amount of these compounds. *Cistus* is an aromatic plant highly valued for its functional properties, among which the content of phenolic compounds is remarkable [23]. *Quercus ilex* was found to have the most phenolic compounds among those extracted from the leaves of native trees. There are few studies on this plant, but some authors have also shown that it possesses a large amount of phenolic compounds. Boy et al. [4] demonstrated that *Quercus ilex* leaf constitutes a rich source of bioactive compounds, especially phenolic compounds, with functional properties. Sánchez Gutiérrez et al. [24] evaluated for the first time the influence of in vitro gastrointestinal digestion on the bioaccessibility and bioactivity of phenolic compounds in the ground leaf and leaf powder extract of this plant, demonstrating important health effects.

Table 2 shows the profile of the compounds extracted from the different plants. It can be observed that there were significant differences regarding the phenolic compound composition of the different plants. Compounds identified in *Cistus albidus* were not found in the other three samples. These compounds were the caffeic acid, eriodictyol, dihydroquercetin, eriodictyol, naringenin, esculetin-O-glucoside, chlorogenic acid and kaempferol diglycoside. Among these compounds, caffeic acid and chlorogenic acid are phenolic compounds of great importance for their antioxidant activity and are frequently identified in plant extracts and kaempferol as a flavonoid [25,26].

A larger variety of compounds were detected in *Quercus ilex*, with 12 compounds, and *Rubus ulmifolius*, with 11 compounds, many of them common as dihydroquercetin, myricetin, cynnamic acid, ligstroside glucuronide, 5-O-Caffeoylquinic acid and 5.7-dihydroxy-3′.4′-dimethoxyflavanone. However, coumaric acid and protocatechuic acid were only detected in *Quercus ilex* extracts and kaempferol in *Rubus*, *Ulmus* and *Cistus* extracts. Coumaric acid is an important phenolic compound for its antioxidant activity and is common in plants, but kaempferol is less common [25]. Phenolics having antioxidant activity are mainly phenolic acids and flavonoids. Considerable variation in phenolic compounds has been found in different plant species. Furthermore, due to the great diversity and complexity of natural mixtures of phenolic compounds in plant extracts, only the major groups are usually identified.

Many studies have shown a strong correlation between the content and composition of phenolic compounds and antioxidant and antimicrobial activity [27].

### 3.2. Activity of the Plants Extracts against Growth and Aflatoxin Production of the Two Micotoxigenic Aspergillus flavus Strains

The plant extracts were tested for their influence on the growth of two *Aspergillus flavus* strains and their effect on aflatoxin production. For this purpose, the concentrations of the extracts were adjusted at 250 and 150 mg/L of total phenolic compounds. This assay was carried out during 11 days of incubation at 25 °C. Figure 1 shows the final growth of strains Cq103 and Cq8 in a PDA medium in the presence of 250 mg/L of the four plant extracts used. Control was performed without the addition of any extract. It can be observed that at the end of the study, *A. flavus* grew with a similar growth rate in the presence of tested extracts except for the *Ulmus* sp. extract, where slower growth was observed in both strains. The final growths were between 8.6 and 5.2 cm in diameter at 11 days for strain CQ103 and between 8.4 and 5.4 cm for strain CQ8.

The results of the growth rate at different days of sampling and with both concentrations of each extract are shown in Table 3.

*A. flavus* Cq103 grew at a rate between 7 and 5.8 mm/day with no significant differences detected between the control and the samples with 150 mg/L and 250 mg/L of the extracts of *Quercus ilex*, *Rubus ulmifolius* and *Cistus albidus* plants. However, at 250 mg/L of *Ulmus* sp. extract, significant differences were detected with respect to the control in the growth rate, which was 4.75 mm/day. Similar results were observed for the study with strain Cq8.

Different studies have observed inhibition of the growth of several types of hazardous molds in food in the presence of plant extracts; however, in our study, this effect on growth was not as marked. Abbaszadeh et al. [28] evaluated the antifungal efficacy of plant phenolic compounds, and the results indicated that all compounds were effective to varying degrees against the growth of different molds. The highest inhibitory activity of pure thymol, carvacrol, eugenol and menthol compounds were found for *Cladosporium* sp. between 100 and 350 μg/mL and *Aspergillus* sp. at 100 and 125 μg/mL. Faustino et al. [29] studied the effects of phenolic extracts obtained from *Calendula* L., rich in hydroxycinnamic acid and flavonoid derivatives, observing significant activity against *Microsporum* strains. Thus, other authors also observed that the inhibition of the growth of various types of harmful molds in food was different when they were in the presence of plant extracts.

Table 4 shows the effect of the selected plant extracts on the production of aflatoxins AFB_1_ and AFB_2_ by *A. flavus* Cq103 and Cq8. In this case, it was observed that while there were no significant differences in growth for aflatoxin production, there were differences with respect to the control. All extracts produced inhibition in terms of aflatoxin production, although to different extents.

In general, AFB_1_ production was higher than AFB_2_ for both strains, and strain Cq103 was less aflatoxin-producing at 25 °C than strain Cq8. In addition, the extracts had higher activity concentrated at 250 mg/L than those at 150 mg/L.

*Quercus ilex* produced the greatest inhibition, followed by *Ulmus* sp. On the other hand, the extract that had the least effect on aflatoxin production was that of *Rubus ulmifolius*.

Strain Cq103 was also inhibited in aflatoxin production by *Quercus ilex* and *Ulmus* sp. extracts, but Cq8 was more inhibited by *Quercus ilex* extracts.

The extracts tested have a different composition of phenolic compounds, as presented above. This may be the cause of the different activity against aflatoxin production by *A. flavus* strains. Different studies on the influence of phenolic compounds on mold growth have been published. However, there are few studies on the influence of aflatoxin production.

The results of the relative expression of *A. flavus* genes in the presence of plant extracts are shown in Table 5. The *aflR* gene of *A. flavus* was activated on expression in the presence of *Rubus ulmifolius* extract for strain Cq8. However, no significant differences were observed. No significant changes were seen in the expression of this regulatory gene in any of its forms.

These results are not as expected since the extracts significantly decreased the production of AFB_1_ and AFB_2_ by both *A. flavus* strains. This inhibition is not observed in the expression of the regulatory gene of this biosynthetic pathway. Previous studies have shown that *aflR* gene expression and AFB_1_ synthesis are closely related [18,30]. This may be because both gene expression and aflatoxin B_1_ and B_2_ production were determined on the same incubation day, at the end of the assay, 11 days. Different studies have shown that the expression of genes involved in aflatoxin biosynthesis is cyclic [20], so there may be a lag between both parameters. Other research has concluded that, although the *aflR* gene has been described as the key gene in aflatoxin synthesis, sometimes there is no relationship between *aflR* gene expression and toxin production [31] depending on the environmental and nutritional conditions tested. On the other hand, the fact that the tested strains are capable of producing more than one mycotoxin makes it more complicated to correlate the expression of a gene with the production of a specific mycotoxin since the mold activates several mechanisms to achieve the synthesis of both secondary metabolites [32].

### 3.3. Multivariate Analysis of the Parameters Studied

Principal component analysis was carried out for phenolic compound content, growth of *A. flavus* strains, aflatoxin production and compounds identified in each of the extracts of the different plants studied to provide an interpretable understanding of the inhibition of aflatoxin production with the compounds identified in each extract.

Figure 2 shows the two-way loadings and score plots, where PC3 was plotted against PC1, explaining 67.37% of the total variance. According to Figure 2, aflatoxin productions were explained by the positive axis of CP3 and were also related to extracts of *R. ulmifolius* and its compounds. Extracts of *Quercus ilex* and *Ulmus* sp. were located on the negative axis of CP3, correlating with the lowest values of aflatoxin production. These were mainly compounds dihydroquercetin, coumaric acid, eriodictyol, protocatechuic acid and cinnamic acid from *Quercus ilex* and *Ulmus* sp. These compounds have been identified in aromatic and medicinal plants with important functional activities, primarily antioxidant activity [24,25,26].

## 4. Conclusions

The results obtained in the present study are of great importance to promote the use of new natural antifungal compounds that inhibit the production of aflatoxins and that have been obtained from the natural resources of the Dehesa, fomenting respect for the environment by avoiding the use of chemical fungicides.

Extracts obtained from plants of the *Quercus ilex* genus were very useful in inhibiting the production of aflatoxin B_1_ and B_2_ produced by two strains of *A. flavus*. This extract was one of the extracts in which a greater number of different compounds were identified and also had a higher concentration of phenolic compounds than *Ulmus* sp.

Furthermore, in this study, it has been demonstrated that the compound composition of the extracts is of vital importance for antifungal activity. In this sense, the compounds identified from the leaves of *Quercus ilex* and *Ulmus* sp. are the ones that may be the most effective to use as potential antifungals.

The gene expression studies involved in the aflatoxin synthesis pathway to determine the production were not effective as they were performed at the end of the trial, and, probably, these genes were expressed earlier.

Consequently, the results obtained are very promising, but future studies are needed to determine whether pure phenolic compounds are the only ones responsible for the antifungal activity of *Quercus ilex* and *Ulmus* sp. and to verify the mechanism of action of these compounds.

## Figures and Tables

**Figure 1 foods-12-01821-f001:**
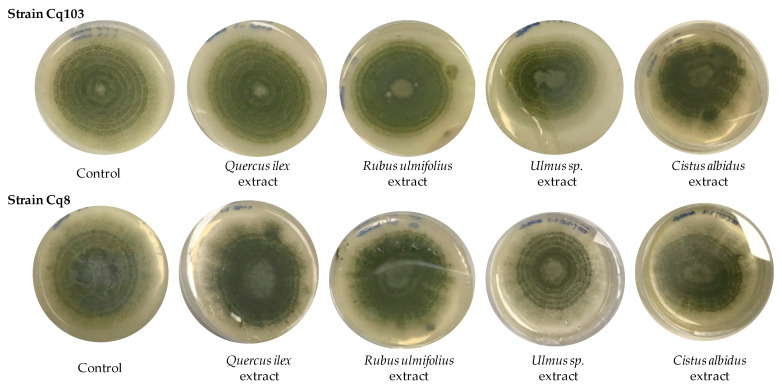
Growth of *A. flavus* strains Cq103 and Cq8 in PDA medium in the presence of 250 mg/L of the four plant extracts and a control without any extract.

**Figure 2 foods-12-01821-f002:**
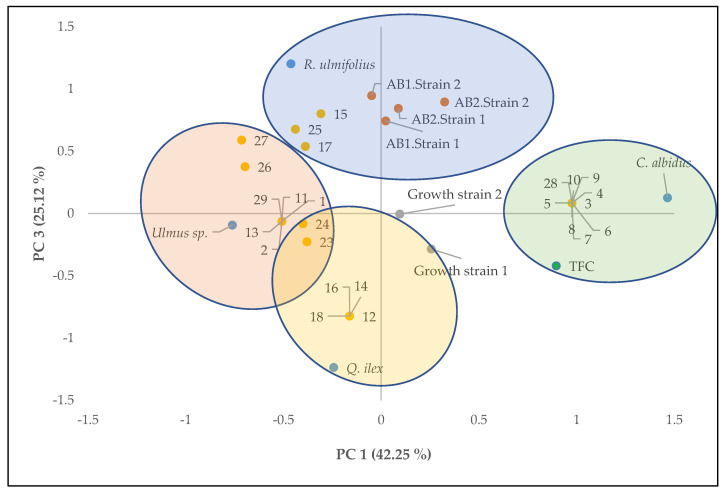
Principal component analysis of the analytical results of the plant extracts (*Quercus ilex*, *Rubus ulmifolius*, *Ulmus* sp. and *Cistus albidus*); total phenolic compounds (TFC); Growth of strain 1 (Cq103) and strain 2 (Cq8); aflatoxin B_1_ and B_2_ production of strains 1 and 2 (AB_1_ strain 1, AB_2_ strain 1, AB_1_ strain 2 and AB_2_ strain 2) at 250 mg/L concentration of the extracts and the phenolic compounds identified (Table 2).

**Table 1 foods-12-01821-t001:** Total phenolic compounds of plant extracts expressed as mg gallic acid equivalents (GAE)/100 g fresh plant.

Extract	Total Phenolic Compounds		
	Mean		SD ^1^
*Quercus ilex*	527	±	72.12 ^b^
*Rubus ulmifolius*	257	±	45.25 ^c^
*Ulmus* sp.	253	±	11.31 ^c^
*Cistus albidus*	703	±	7.07 ^a^

^1^ SD, standard deviation. ^a,b,c^ values with different superscripts are significantly different (*p* ≤ 0.05) between samples.

**Table 2 foods-12-01821-t002:** Identification and quantification in arbitrary area units of bioactive compounds from plant extracts analyzed by HPLC-UV-ESI-MS/MS.

Peak	Rt (min)	[M-H]^−^ (*m*/*z*)	MS/MS (*m*/*z*)	Compound Identified *	*Quercus ilex*	*Rubus ulmifolius*	*Ulmus* sp.	*Cistus albidus*
1	3.020	116	277; 365	L-Valine	0.00 ^b^	0.00 ^b^	47.24 ^a^	0.00 ^b^
2	4.167	123	144	Orcinol	0.00 ^b^	0.00 ^b^	36.63 ^a^	0.00 ^b^
3	10.529	179	133; 135; 180	Caffeic acid	0.00 ^b^	0.00 ^b^	0.00 ^b^	44,979.47 ^a^
4	11.026	179	105; 133; 180; 395	Caffeic acid	0.00 ^b^	0.00 ^b^	0.00 ^b^	5876.58 ^a^
5	11.441	287	449; 611	Eriodictyol	0.00 ^b^	0.00 ^b^	0.00 ^b^	910.52 ^a^
6	11.69	303		Dihydroquercetin	0.00 ^b^	0.00 ^b^	0.00 ^b^	384.87 ^a^
7	11.839	287		Eriodictyol	0.00 ^b^	0.00 ^b^	0.00 ^b^	6107.77 ^a^
8	12.204	271	287	Naringenin	0.00 ^b^	0.00 ^b^	0.00 ^b^	1926.97 ^a^
9	12.403	339	165; 323; 501	Esculetin-O-glucoside	0.00 ^b^	0.00 ^b^	0.00 ^b^	419.97 ^a^
10	12.850	353	127; 659	Chlorogenic acid	0.00 ^b^	0.00 ^b^	0.00 ^b^	5448.26 ^a^
11	13.603	303	123; 139	Dihydroquercetin	0.00 ^b^	0.00 ^b^	7517.37 ^a^	0.00 ^b^
12	13.664	163	164	Coumaric acid	473.95 ^a^	0.00 ^b^	0.00 ^b^	0.00 ^b^
13	13.835	287		Eriodictyol	0.00 ^b^	0.00 ^b^	5914.22 ^a^	0.00 ^b^
14	13.846	153	171	Protocatechuic acid	255.62 ^a^	0.00 ^b^	0.00 ^b^	0.00 ^b^
15	13.969	303		Dihydroquercetin	0.00 ^b^	761.21 ^a^	0.00 ^b^	0.00 ^b^
16	14.062	303		Dihydroquercetin	7605.68 ^a^	0.00 ^b^	0.00 ^b^	0.00 ^b^
17	14.317	317	318	Myricetin	309.19 ^b^	867.96 ^a^	0.00 ^c^	0.00 ^c^
18	14.443	163	163	Coumaric acid	1890.40 ^a^	0.00 ^b^	0.00 ^b^	0.00 ^b^
19	15.173	147	148	Cynnamic acid	7496.65 ^a^	3785.26 ^b^	0.00 ^c^	0.00 ^c^
20	16.208	147	148	Cynnamic acid	1726.29 ^b^	4078.55 ^a^	0.00 ^c^	0.00 ^c^
21	17.550	147	148	Cynnamic acid	784.53 ^b^	3021.26 ^a^	0.00 ^c^	0.00 ^c^
22	17.750	147	148; 283; 847	Cynnamic acid	806.54 ^a^	870.59 ^a^	0.00 ^b^	0.00 ^b^
23	19.862	537	261; 511	Ligstroside glucuronide	660.18 ^a^	487.32 ^a^	0.00 ^b^	0.00 ^b^
24	19.945	353	354	5-O-Caffeoylquinic acid	653.41 ^a^	597.34 ^a^	0.00 ^b^	0.00 ^b^
25	20.138	315	316	5.7-Dihydroxy-3′.4′-dimethoxyflavanone	362.77 ^b^	1663.85 ^a^	181.54 ^b^	0.00 ^c^
26	21.149	593	533; 534; 594	Kaempferol 3-O-rutinoside	0.00 ^c^	8806.99 ^b^	16,605.12 ^a^	0.00 ^c^
27	23.173	533	534; 593; 594	Kaempferol 3-O-malonyl-glucoside	0.00 ^c^	1163.51 ^b^	1323.69 ^a^	0.00 ^c^
28	23.497	593	533	Kaempferol diglycoside	0.00 ^b^	0.00 ^b^	0.00 ^b^	846.83 ^a^
29	24.536	635	575; 576; 636	Trigalloyl-hexoside	0.00 ^b^	0.00 ^b^	200.31 ^a^	0.00 ^b^

* MassBank ^a,b,c^ values with different superscripts are significantly different between plant extracts (Tukey’s test; *p* < 0.05).

**Table 3 foods-12-01821-t003:** Growth rate (mm/day) of *A. flavus* strains Cq103 and Cq8 obtained during 11 days of incubation at 25 °C.

Extract	Concentration (mg/L)	Cq103 Strain			Cq8 Strain		
		Mean		SD ^1^	Mean		SD
Control	0	7.04	±	0.51 ^a,b^	7.69	±	0.21 ^a^
*Quercus ilex*	150	6.40	±	0.07 ^a,b^	7.19	±	0.13 ^a^
	250	6.63	±	0.77 ^a,b^	6.94	±	0.33 ^a,b^
*Rubus ulmifolius*	150	6.50	±	0.20 ^a,b^	7.01	±	0.48 ^a,b^
	250	6.01	±	0.19 ^a,b^	6.87	±	0.07 ^a,b^
*Ulmus* sp.	150	7.88	±	0.01 ^a^	7.05	±	0.03 ^a,b^
	250	4.75	±	0.66 ^b^	4.96	±	0.58 ^c^
*Cistus albidus*	150	6.93	±	0.69 ^a,b^	7.17	±	0.50 ^a^
	250	5.87	±	0.34 ^a,b^	6.07	±	0.51 ^b^

^1^ SD, standard deviation. ^a,b,c^ values with different superscripts are significantly different (*p* ≤ 0.05) between samples for each column.

**Table 4 foods-12-01821-t004:** Aflatoxin B_1_ and B_2_ (ppb) of strains Cq103 and Cq8 of *A. flavus* determined in the samples with the different extracts at the end of the assay at 25 °C.

Extract	Concentration (mg/L)	Cq103 *A. flavus* Strain						Cq8 *A. flavus* Strain					
		Aflatoxin B_1_ (ppb)			Aflatoxin B_2_ (ppb)			Aflatoxin B_1_ (ppb)			Aflatoxin B_2_ (ppb)		
		Mean		SD ^1^	Mean		SD	Mean		SD	Mean		SD
Control	0	210.82	±	24.74 ^a^	14.12	±	0.83 ^a^	621.31	±	164.20 ^a^	37.05	±	13.99 ^a,b^
*Quercus ilex*	150	64.37	±	11.71 ^b^	2.74	±	0.27 ^b^	199.00	±	12.73 ^b,c,d^	8.07	±	2.22 ^b^
	250	53.22	±	6.90 ^b^	3.33	±	0.48 ^b^	105.06	±	21.26 ^d^	3.51	±	1.19 ^b^
*Rubus ulmifolius*	150	106.01	±	24.37 ^b^	8.93	±	0.27 ^b^	446.05	±	60.43 ^a,b^	60.86	±	1.53 ^a^
	250	96.01	±	74.37 ^b^	6.93	±	5.27 ^b^	411.05	±	30.43 ^a,b,c^	30.86	±	1.53 ^b^
*Ulmus* sp.	150	69.86	±	13.11 ^b^	4.42	±	0.90 ^b^	217.38	±	42.71 ^b,c,d^	9.51	±	1.77 ^b^
	250	39.66	±	24.05 ^b^	2.99	±	2.00 ^b^	163.44	±	74.01 ^c,d^	6.44	±	4.15 ^b^
*Cistus albidus*	150	94.00	±	19.30 ^b^	6.97	±	2.23 ^b^	402.93	±	76.71 ^b,c^	65.73	±	15.21 ^a^
	250	60.84	±	12.11 ^b^	4.61	±	1.20 ^b^	218.68	±	25.26 ^b,c,d^	22.31	±	4.77 ^b^

^1^ SD, standard deviation, ^a,b,c,d^ values with different superscripts are significantly different (*p* ≤ 0.05) between samples for each column.

**Table 5 foods-12-01821-t005:** Relative gene expression log2 of the *aflR* gene of *A. flavus* strains Cq103 and Cq8 determined in the samples with the different extracts at the end of the assay at 25 °C.

Extract	Concentration (mg/L)	Cq103 *A. flavus* Strain			Cq8 *A. flavus* Strain		
		Mean		SD ^1^	Mean		SD
Control	0	0.00	±	0.000	0.00	±	0.000
*Quercus ilex*	150	−0.46	±	0.337	−0.12	±	1.061
	250	−3.04	±	0.001	−0.26	±	0.742
*Rubus ulmifolius*	150	−2.37	±	0.004	0.21	±	0.531
	250	−2.49	±	0.004	0.21	±	0.531
*Ulmus* sp.	150	−0.40	±	0.519	−2.00	±	0.012
	250	−0.58	±	0.321	0.00	±	1.087
*Cistus albidus*	150	−0.27	±	0.593	−2.17	±	0.000
	250	−0.29	±	0.647	−1.79	±	0.023

^1^ SD, standard deviation.

## Data Availability

Data is contained within the article.

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
