# Peer review of "Antifungal Effect of Autochthonous Aromatic Plant Extracts on Two Mycotoxigenic Strains of Aspergillus flavus"

_foods, 2023, doi:10.3390/foods12091821_

Round 1
Reviewer 1 Report
Dear Authors,
After reviewing your paper, I feel that certain corrections need to be made. The paper deals with an extremely current topic that is being dealt with by an increasing number of researchers. Your work has a certain potential, and I leave you suggestions to further improve it.
I believe that the introductory part is not directed in the right way in relation to the goal and framework of this paper. Try to present the literature review in a more adequate way, emphasizing the importance of released microorganisms, mycotoxins, impact on people, etc. The last few papers I had a chance to review were about the effects of mycotoxins on babies and children, so I'll leave you space to mention that as well. This will give you a better chance of being cited in the future.
In the Materials and methods section, I would ask you to add more information about plant material, growing and storage conditions, as this is one of the triggers for the development of molds and their products.
In some parts of the paper is very difficult to understand text, please check the whole paper and correct meaning of some sentences.
Author Response
After reviewing your paper, I feel that certain corrections need to be made. The paper deals with an extremely current topic that is being dealt with by an increasing number of researchers. Your work has a certain potential, and I leave you suggestions to further improve it.
Thank you for your comments and suggestions, which we believe are very helpful and increase the quality of this paper.
I believe that the introductory part is not directed in the right way in relation to the goal and framework of this paper. Try to present the literature review in a more adequate way, emphasizing the importance of released microorganisms, mycotoxins, impact on people, etc. The last few papers I had a chance to review were about the effects of mycotoxins on babies and children, so I'll leave you space to mention that as well. This will give you a better chance of being cited in the future.
Thank you for your comments and suggestions. The introductory part has been modified to emphasize the importance of microorganisms, mycotoxins, the impact on people, etc. The effects of mycotoxins on babies and children has also been highlighted.
In the Materials and methods section, I would ask you to add more information about plant material, growing and storage conditions, as this is one of the triggers for the development of molds and their products.
Detailed information on the place, date, and storage conditions has been added: ” provided by a company located in a Dehesa area of Peraleda de la Mata municipality, Cáceres, Extremadura. They were collected in February 2021. All samples were packed in vacuum bags and stored at 4° C until use.”
Comments on the Quality of English Language: In some parts of the paper is very difficult to understand text, please check the whole paper and correct meaning of some sentences.
Several sentences has been checked and clarified.
Reviewer 2 Report
Dear Authors,
Dehesa is a region in Spain that has been declared a UNESCO Biosphere Reserve. The Authors undertook the task of searching for alternatives to existing antifungal agents. The aim of this study was to find these compounds among natural resources obtained from Quercus ilex (Holm oak), Ulmus sp. (Elm), Rubus ulmifolius (Blackberry) and the flowers of Cistus albidus (White rockrose).
The manuscript is thoughtful, meticulously prepared. It contains minor errors that need to be improved:
The Extremadura Dehesa is characterized as a very diverse ecosystem where a great heterogeneity of medicinal plants can be found, in addition to aromatic plants [4], among which Quercus ilex can be included, highlighted for its anti-inflammatory, antibacterial, hepatoprotective, antidiabetic , anticancer, gastroprotective, antioxidant and cytotoxic properties [5], Ulmus sp., Rubus ulmifolius and Cistus albidus, from which even vitamins (ascorbic acid), reducing sugars (glucose and fructose), polyunsaturated fatty acids and eicosadienoic acids have been obtained [ 6]. – sentence too long, incomprehensible to the reader. Please separate and increase the readability and clarity.
The aflR gene name should be in italics. Please include this throughout the Manuscript
Table 2 should be a paragraph above, as it is mentioned there for the first time.
Periods in which plants were collected for research should be added. Often their composition (as well as the ratios of individual compounds), and thus also the properties, depend on the season/month in which they are collected.
When describing the purpose of research, Dehesa should be capitalized.
\Best regards
Author Response
The authors appreciate the reviewer's suggestions, which are very useful and improve the quality of the work.
Dehesa is a region in Spain that has been declared a UNESCO Biosphere Reserve. The Authors undertook the task of searching for alternatives to existing antifungal agents. The aim of this study was to find these compounds among natural resources obtained from Quercus ilex (Holm oak), Ulmus sp. (Elm), Rubus ulmifolius (Blackberry) and the flowers of Cistus albidus (White rockrose).
Thank you, the sentence La Dehesa is a Spanish region declared a Biosphere Reserve by UNESCO, has been included.
The manuscript is thoughtful, meticulously prepared. It contains minor errors that need to be improved:
The Extremadura Dehesa is characterized as a very diverse ecosystem where a great heterogeneity of medicinal plants can be found, in addition to aromatic plants [4], among which Quercus ilex can be included, highlighted for its anti-inflammatory, antibacterial, hepatoprotective, antidiabetic , anticancer, gastroprotective, antioxidant and cytotoxic properties [5], Ulmus sp., Rubus ulmifolius and Cistus albidus, from which even vitamins (ascorbic acid), reducing sugars (glucose and fructose), polyunsaturated fatty acids and eicosadienoic acids have been obtained [ 6]. – sentence too long, incomprehensible to the reader. Please separate and increase the readability and clarity.
Paragraph has been separated and clarified
The aflR gene name should be in italics. Please include this throughout the Manuscript
It has been changed.
Table 2 should be a paragraph above, as it is mentioned there for the first time.
It has been included.
Periods in which plants were collected for research should be added. Often their composition (as well as the ratios of individual compounds), and thus also the properties, depend on the season/month in which they are collected.
Detailed information on the place, date, and storage conditions has been provided.
When describing the purpose of research, Dehesa should be capitalized.
It has been changed